Comparative transcriptome profiling suggests the role of phytohormones in leaf stalk-stem angle in melon (Cucumis melo L.)

Mao Jiancai
Wang Haojie
Li Junhua
Yang Junyan
Zhang Yongbing hamimelon@xaas.ac.cn
Wu Haibo wuhaibo@xaas.ac.cn
Hami Melon Research Center, Xinjiang Academy of Agricultural Sciences , Urumqi , China
Abd El-Moneim Diaa
Electronic publication date: 2024 Nov 18
Publication date: 2024
Volume: 12
Electronic Location ID: e18467
Received 2024 Aug 19; Accepted 2024 Oct 15
Copyright: ©2024 Mao et al.
Copyright year: 2024
Copyright holder: Mao et al.
License: This is an open access article distributed under the terms of the Creative Commons Attribution License, which permits unrestricted use, distribution, reproduction and adaptation in any medium and for any purpose provided that it is properly attributed. For attribution, the original author(s), title, publication source (PeerJ) and either DOI or URL of the article must be cited.
License URL: https://creativecommons.org/licenses/by/4.0/

Keywords: Melon, Leaf stalk-stem angle, Comparative transcriptome, Phytohormone

Funding: Public Welfare basic scientific research funds of Xinjiang Academy of Agricultural Sciences Grant No. KY2023015 Tianshan Talent Training Program Grant No. 2023TSYCLJ0015 XinJiang Agriculture Research System XJARS-6 Xinjiang-Central Asia Uzbekistan Agricultural Technology Sharing ZTE Construction Project 2022E01035 This research was funded by the Public Welfare basic scientific research funds of the Xinjiang Academy of Agricultural Sciences (Grant No. KY2023015), the Tianshan Talent Training Program (Grant No. 2023TSYCLJ0015), the Xinjiang Agriculture Research System (XJARS-6) and Xinjiang-Central Asia Uzbekistan Agricultural Technology Sharing ZTE Construction Project (2022E01035). The funders had no role in study design, data collection and analysis, decision to publish, or preparation of the manuscript.

==============================
Leaf stalk-stem angle is an important agronomic trait influencing melon architecture, photosynthetic efficiency, and crop yield. However, the mechanisms governing leaf stalk-stem angle, particularly in melon, are not well understood. In this study, we explored the comparative transcriptome in the expanded architecture line Y164 and the compact plant architecture line Z151 at 30 days after pollination. Phytohormones were measured at the leaf stalk-angle site at the same time in these two lines using liquid chromatography (LC) tandem mass spectrometry (MS) (LC-MS/MS). The phytohormones and transcriptomes were jointly analyzed. Differential hormone profiling revealed that the levels of 1-aminocyclopropane-1-carboxylate (ACC) and 12-oxophytodienoic acid (OPDA) in the large-angled line Y164 were significantly higher than those in the small-angled line Z151. These differences were quantified as 2.1- and 2.8-fold increases, respectively. Conversely, the content of isopentenyl adenosine (IPA) was significantly elevated in Z151, with a 3.8-fold higher concentration relative to Y164. Transcriptome analysis identified a total of 1709 differently expressed genes (DEGs), with a predominant enrichment in the Kyoto Encyclopedia of Genes and Genomes (KEGG) pathways related to photosynthesis and plant hormone signal transduction. Similarly, photosynthesis and the hormone metabolic process were predominantly enriched in the biological process of Gene Ontology (GO) terms. Further integration of transcriptome and hormone analyses substantiated the close relationship between melon leaf stalk-stem angle and phytohormones, especially ACC, OPDA and IPA. Selected DEGs from phytohormone signal transduction were validated. Detailed analysis of DEGs highlighted the potential role of genes such as GH3s (LOC103490488, LOC103490483), SUARs (LOC107991561, LOC103497281 and LOC103489067), ARFs (LOC103503893, LOC103493078) and five genes in abscisic acid pathway. In summary, our findings strongly suggest a direct correlation between phytohormones and the leaf stalk-stem angles in melon.

Introduction

Plant architecture significantly influences light acceptance, photosynthetic production, and nutrient partitioning, playing an important role in crop yield, product quality, and cultivation management (Tong & Chu, 2018; Jiao et al., 2010). The unique role of plant architecture has captured the attention of breeders, as evidenced by the “green revolution” in grass crops, which underscores the immense potential of plant architecture breeding.

In recent years, melon cultivation has seen substantial growth, particularly in greenhouse facilities that predominantly employ hanging cultivation. The effectiveness of greenhouse cultivation of melon is closely related to the plant architecture. In cases where melon plants exhibit extensive expansion of architecture, leaves tend to shade each other, impacting ventilation. Additionally, as leaves and petioles increase, there is a heightened risk of breakage, susceptibility to disease and pests, and an increase in management challenges. On the contrary, melons with a compact plant architecture optimize land and light utilization, improve ventilation and light transmission, reduce disease and pest occurrence, and allow for higher planting density and unit area yield (Xiong et al., 2016). The leaf stalk-stem angle of melons, denoting the angle between the main vine and petiole, is a pivotal factor in melon plant architecture. Studies indicate that a smaller leaf stalk-stem angle results in better light distribution in the middle and lower part of the melon population, particularly advantageous for greenhouse cultivation (Knavel, 1990; Fukino et al., 2012; Yang et al., 2020). Therefore, in the current melon breeding research, the leaf stalk-stem angle trait has gained increasing attention. The identification of quantitative trait loci (QTLs) and genes associated with this trait, along with an elucidation of its molecular mechanisms, forms the foundation for genetically improving the leaf stalk-stem angle in melons. Despite several studies indicating the major gene control of melon leaf stalk-stem angle, environmental factors exert a significant influence on this trait (Xiong et al., 2016). However, the specific gene responsible for this trait remains elusive.

Leaf stalk-stem angle represents a typical complex quantitative trait controlled by multiple genes. Its exploration is relatively limited in Cucurbitaceae crops compared to Gramineae crops, where the study of a similar trait, leaf angle, is more extensive. In Gramineae crops such as maize (Wang et al., 2022; Cao et al., 2020; Ren et al., 2020), wheat (Liu et al., 2018; Liu et al., 2019), rice (Huang et al., 2023; Guo et al., 2021), and others (Yang et al., 2023), numerous genes influencing leaf angle have been identified. Many of these genes are associated with phytohormones, establishing a genetic foundation for enhancing crop strains. Most of the current studies have shown that members of the IGT/LAZY gene family mediate the regulation of phototropic and geotropic responses in branching angle. Related genes, such as LAZY1 (LZY1) and LZY2, LZY4, sense geotropic signals to alter the local auxin gradient. This asymmetric auxin distribution induces two WUSCHEL-related homeobox domain genes WOX6 and WOX11, ultimately affecting the branch/tiller angle (Hill & Hollender, 2019). In addition, phytohormones, such as brassinosteroid (BR) and gibberellic acid (GA), are well-known for their essential roles in regulating leaf angle (Huang et al., 2021; Huang et al., 2023; Tong & Chu, 2018; Luo et al., 2016). These phytohormones stimulate polysaccharide synthesis, increase cell wall elasticity, and promote the synthesis of new cell wall components and cell elongation. Consequently, they have a profound impact on plant architecture, influencing factors such as leaf angle (Luo et al., 2016).

BRs are ubiquitously distributed in all growing tissues of higher-plants and play a vital role in promoting cell elongation (Liang et al., 2014). BR biosynthesis and metabolism determines BR homeostasis, and several studies highlighted that mutants with impaired BR synthesis often exhibit altered leaf angle (Liang et al., 2014). In maize, the BR biosynthesis-related gene Nana plant 2 (NA2) encodes a sterol reductase that converts 24-methylenecholesterol to campesterol. This gene is notably expressed in the ligule region of developing leaves, and mutations in NA2 cause defects in BR synthesis, resulting in an increased leaf angle (Best et al., 2016). Similarly, mutations in ZmDWF4, a key player in the BR biosynthesis pathway, limit BR synthesis, causing a reduction in maize leaf angle and overall plant height (Liu et al., 2007). Other studies in maize have identified genes related to BR biosynthesis, such as BRD1, LG1, and ZmBRII, where mutations induce changes in leaf angle (Makarevitch et al., 2012; Huang et al., 2023; Kir et al., 2015; Tian et al., 2019). Additionally, studies in rice, wheat and other crops have demonstrated that BRs control leaf angle by regulating the asymmetric elongation of adaxial cells (Liu et al., 2018; Gao et al., 2018; Huang et al., 2023; Sakakibara, 2006). Collectively, these findings underscore the crucial role of BRs in governing leaf angle in various crops.

Auxin, another crucial plant phytohormone, regulates leaf angle. Light has been shown to control leaf auricular growth and leaf angle size by influencing polar auxin transport (Fellner et al., 2003). In maize, ZmPGP1 encodes an auxin efflux carrier P-glycoprotein involved in the polar auxin transport. A deletion mutation in ZmPGP1 significantly reduces plant height, increases stem thickness, reduces leaf angle, and enhances resistance to stunting (Wei et al., 2018). Rice exhibits control over leaf angle size through the regulation of auxin response factors OsARF6 and OsARF17 (Huang et al., 2021). In soybean, the auxin transporter PINFORMED1 (GmPIN1) mediates asymmetric distribution of auxin, thereby influencing the asymmetric expansion of cells in the soybean petiole, regulating the leaf angle and compacting plant structure (Huang et al., 2021; Zhang et al., 2022). In addition, ethylene plays an important role in leaf angle formation and development. In maize, the key gene ZmACS7 for ethylene biosynthesis encodes 1-aminocyclopropane-1-carboxylate (ACC) synthase. Alterations in its C-terminal leads to ACC and ethylene accumulation, promoting the longitudinal elongation of leaf auricles and ultimately increasing leaf angle (Li et al., 2020). It is worth noting that phytohormones often interact with each other and do not solely regulate plant traits such as leaf angle. For example, both BR and GA stimulate cell growth and increase leaf angle. The jade rice BR synthesis-deficient mutants na2-1 and na1-1, along with the GA synthesis-deficient mutant d1, exhibit nearly identical leaf angle. After exogenous GA application, the leaf angle becomes erect and decreases, suggesting that the BR mutant responds to GA, offering a potential avenue for regulating plant phenotypic traits, such as leaf angle (Best et al., 2016). In addition, other factors involved in the crosstalk of phytohormone signaling pathways have been identified as regulators of plant leaf angle (Best et al., 2016). Despite extensive research on leaf angle traits in various crops and the elucidation of related regulatory mechanisms, there is currently no reported research on this aspect in melon.

In this study, we selected the compact architecture line Z151 and the expanded architecture melon Y164 for examination. Leaf stalk-stem angle samples were collected 30 days after melon pollination for RNA-seq and phytohormone analyses. For the first time, we scrutinized the differential phytohormone profiles between the two lines, shedding light on the pivotal roles of phytohormones in shaping the leaf stalk-stem angle in melon. Subsequently, we assessed transcriptome variances between the two lines, identifying differentially expressed genes and elucidating the major pathways implicated in the response. Finally, the role of phytohormones in regulating melon leaf stalk-stem angle was further clarified through a combined transcriptome-hormone analysis. This study provides empirical evidence for the gene localization of melon stalk-stem angle and provides valuable insights into the molecular mechanisms underlying phytohormone regulation of leaf stalk-stem angle in melon.

Materials and Methods

Plant materials and sampling

The compact architecture melon Z151 and the expanded architecture melon Y164 were preserved in the Cantaloupe Research Center of Xinjiang Academy of Agricultural Sciences.Z151 and Y164 are two lines from the RIL population and were observed to be identical in all other traits except leaf stalk-stem angle, the production process is shown in Fig. S1. Planting commenced in March 2023 at Kashgar (35°20′N, 73°20′E, 1264 m), with sampling conducted 30 days after pollination. The sampling site is shown in Fig. 1, encompassing the 12 sections of the main vine. Samples were transported in dry ice to the laboratory, and stored at −80 °C until further use. Three biological replicates were used for phytohormone quantification and RNA-seq analysis.

Figure 1 Comparison of quantitative indicators of leaf stalk-stem angle in Z151 and Y164 line.

Leaf stalk-stem angle measurement

The leaf-stem angle was measured from the 11th to the 15th node of the plant with a digital angle ruler. The average value derived from these measurements was considered the stalk-stem angle. Recognizing the potential impact of temporary wilting induced by sunlight on the leaf-stem angle, measurements were conducted in the morning to mitigate this effect. To ensure consistency and accuracy, all trait measurements were executed by the same individual, thereby eliminating potential systematic errors and enhancing the reliability of the data.

Phytohormone analysis

Standard configuration

A total of 34 phytohormone standards were accurately weighed to obtain a mixed standard solution solved in methanol (Table S1). Secondary working solutions were prepared through serial dilution using methanol. The standard solutions were stocked under −20 °C for future use.

An appropriate amount of the two internal standards of phytohormones was accurately weighed to prepare a 1mg/mL internal standard single-standard stock solution with methanol. Each single-standard stock solution was diluted with methanol to prepare a solution containing 0.5 µg/mL IAA-D5 and 1 µg /mL ABA-D6 mixed internal standard stock solution, and stored at −20 °C.

A total of 15 µL of the standard secondary working solution and 0.5 µL of the mixed internal standard stock solution were added to the injection bottle with 134.5 µL of sample QC to make the standard working solution for liquid chromatography (LC) tandem mass spectrometry (MS) (LC-MS/MS) analysis.

Metabolite extraction

For metabolite extraction, an appropriate amount of lyophilized sample was weighed and placed in a two mL brown centrifuge tube. Subsequently, one mL of methanol and the accurately prepared mixed internal standard stock solution were added to the tube. The mixture underwent sonication for 10 min and was then transferred to a metal bath for shaking over a 4-hour duration. Afterward, centrifugation was conducted at 12,000 g for 10 min at 4 °C. The entire supernatant was carefully removed post-centrifugation. Following this, 0.5 mL of methanol was added to the remaining residue, and the solution continued to be shaken for an additional 2 h using a metal bath. The resulting extracts were subjected to centrifugation through a 0.22 µm filter membrane and subsequently placed in an injection vial for liquid chromatography (LC) tandem mass spectrometry (MS) (LC-MS/MS) analysis.

The content of sample was calculated using the formula C*V*D*1000/m, where C represents the concentration obtained by substituting the integral peak area ratio of the sample into the standard curve (ng/mL), V is the extractant volume (mL), D donates the dilution factor of the solution, and M represents the weighed sample mass (mg).

Liquid chromatography conditions

UPLC separation was performed on ExionLC UPLC system (AB Sciex, Framingham, MA, USA) equipped with an Acquity UPLC® CSH C18 (1.7 µm, 2.1x150 mm, Waters) column. The temperature of the column was set at 40 °C. The sample injection volume was 2 µL. Eluents consisted of 0.05% formic acid with 2 mM ammonium formate in water (eluent A) and 0.05% formic acid in methanol (eluent B). The flow rate was set at 0.25 mL/min. An elution gradient was performed as follows: 0∼2 min, 10% B; 2∼4 min, 10∼30% B; 4∼19 min, 30∼95% B; 19∼19.10 min, 95∼10% B; 19.10∼22 min, 10% B.

Mass spectrum conditions

The mass spectrum (MS) analysis was performed using an AB Sciex Triple Quadrupole 6500 plus mass spectrometer (AB Sciex) in the multiple reaction monitoring (MRM) mode. The electrospray ionization (ESI) parameters in positive mode were as follows: ion source voltage, 4500 V; ion source temperature, 400 °C; curtain gas, 40 psi; atomization gas, 40 psi and auxiliary gas, 25 psi. The parameters were similar in negative mode, except for ion source voltage which was −4500 V. The parameters of each standard were shown in Table S4.

Transcriptome analyses

Transcriptome sequencing was entrusted to Shanghai Personalbio Technology Co., Ltd. Library quality was assessed using standard metrics. Three biological replications and three technical replications were used in the experiment.

The process for differentially expressed gene screening involves filtering raw data to obtain high-quality clean data. Subsequently, clean data is aligned to the reference genome (https://www.ncbi.nlm.nih.gov/genome/?term=Cucumis+melo), leading to the generation of mapped data. The quality of sequencing libraries is evaluated through tests such as insertion fragment length and randomness. Additionally, the quality of expression analysis, variable splicing analysis, new gene mining, and gene structure optimization is assessed. Expression analysis, variable splicing analysis, new gene mining and gene structure optimization were carried out based on the mapped data. Using the expression levels of genes in different samples or groups, differential expression analysis, functional annotation of differentially expressed genes and functional enrichment were performed.

Combined transcriptome and phytohormone analysis

Correlation analysis and two-way orthogonal PLS (O2PLS) analysis were performed on the results of quantitative assays of phytohormones and transcriptomes. Differentially expressed metabolite and transcript information was extracted, and correlation results were screened and analyzed. Transcripts corresponding to related enzymes were identified based on metabolite information in the KEGG database. This approach allowed for the identification and sorting of the differential trends of differentially expressed metabolites and transcripts with correspondent relationships. Ultimately, pathways common to the analysis of the differential enrichment of the two groups were identified. Additionally, the pathways commonly annotated for differentially metabolized products and genes were sorted out through this comprehensive analysis.

Statistical analysis

The transcriptome assembly and quantification were performed using Stringtie v2.1.3b, and the statistics of the bam comparison results were carried out using Qualimap 2.2.1. For analyzing biological replicates’ variable shear, rMATS 3.1.0 was employed. Heatmaps were generated using R Studio. The analysis of differences with replicates was conducted using DESeq 2 1.26.0, while edgeR 3.28.1 was utilized for the analysis of variance.

Results

Comparison of quantitative indicators of leaf stalk-stem angle in Z151 and Y164

We observed a significant difference in leaf stalk-stem angle between the expanded architecture line Y164 and the compact architecture line Z151 (Fig. 1A). The leaf stalk-stem angle range of Y164, observed from node 11 to 15, spanned between 87.8 and 124 degrees, while that of Z151 ranged from 18.5 and 38.1 degrees, indicating a significant difference (Fig. 1B). Moreover, the average angle for Y164 is 97.6 degrees, contrasting sharply with Z151 average angle of 31.5 degrees, highlighting an extremely significant difference (Fig. 1C).

Quantification of endogenous phytohormones

The phytohormone contents of the two architecture types exhibited substantial variations during the same period of time. Compared to Z151, Y164 demonstrated significantly higher levels of phytohormones, including ACC, SAG, SA and OPDA. Since OPDA is a biosynthetic precursor of JA, JA signaling pathway could possibly play an important role in controlling melon leaf stalk-stem angle. Conversely, other phytohormones, such as IPA, IP, TZR, TZ and ICA, were significantly lower in Y164 when compared to Z151 (Table 1).

Table 1 Phytohormone concentrations in Z151 and Y164 melon varieties.

	IPA (ng/ml)	IP (ng/ml)	TZR (ng/ml)	CZ (ng/ml)	TZ (ng/ml)	Me-IAA (ng/ml)	IAA (ng/ml)	ACC (ng/ml)	ABA (ng/ml)	GA7 (ng/ml)	GA24 (ng/ml)	JA-Ile (ng/ml)	SAG (ng/ml)	GA8 (ng/ml)	SA (ng/ml)	ICA (ng/ml)	ICAId (ng/ml)	OPDA (ng/ml)	
Z151	1.73a	0.05a	0.86a	0.03a	0.14a	0.04a	0.24a	142.04b	3.41a	0.09a	0.10a	0.21a	62.93b	0.26a	17.40b	0.77a	0.69a	3.14b	
Y164	0.45b	0.01b	0.35b	0.03a	0.02b	0.04a	0.25a	298.09a	3.94a	0.15a	0.10a	0.27a	1970.73a	0.29a	44.23a	0.45a	0.41a	8.82a	
Notes.

The data are presented as the mean. Different lowercase letters indicate significance at p < 0.05 level.

IPA isopentenyl adenosine

IP N6-isopentenyladenine

TZR Trans-zeatin Riboside

CZ cis-zeatin

TZ Trans-Zeatin

Me-IAA 2-(1H-indol-3-yl)acetate

IAA indole-3-acetic acid

ACC 1-Aminocyclopropanecarboxylic acid

ABA abscisic acid

GA7 Gibberellin A7

GA24 Gibberellin A24

JA-Ile N-[(-)-jasmonoyl]-(S)-isoleucine

SAG salicylic acid glucoside

GA8 Gibberellin A8

SA salicylic acid

ICA 3-Indoleformic acid

ICAId Indole-3-carboxaldehyde

OPDA 12-oxophytodienoic acid

Figure 2 Differentially accumulated phytohormone screening.

Differentially accumulated phytohormone screening

Our primary objective was to identify relevant differentially accumulated phytohormones based on statistical tests with criteria p-value < 0.05 and VIP > 1 (Kieffer et al., 2016). Subsequent refinement revealed three highly significant differences in phytohormones between the two samples, which are ACC, IPA, and OPDA (Fig. 2A). In Y164, ACC and OPDA were significantly up-regulated, whereas IPA was significantly down-regulated (Figs. 2B, 2C and 2D). These results suggest that the up-regulation of ACC and OPDA may lead to the increase of the leaf-stem angle, while the down-regulation of IPA may lead to the decrease of the angle. This focused screening approach allows us to pinpoint key phytohormone variations, providing valuable insights into the distinct physiological responses of the two plant architectures.

Transcriptomic analysis of different architecture lines

In this analysis, we conducted transcriptome sequencing for six samples, resulting in a total of 85.56 G of clean data. The raw reads obtained ranged from 47.44 to 51.41 million. The effective data volume for each sample varied between 6.84 to 7.43 G, with a Q30 base ranging from 94.60 to 94.95% (Table S1). Following read alignment to the reference genome, each sample demonstrated a genome alignment rate between 95.36 and 97.13% (Table S2).

To delineate differentially expressed genes (DEGs) in the distinct architecture types, we measured gene expression levels and identified DEGs using R packages. Volcano distribution plots illustrating up and down-regulated genes between the two lines are shown in Fig. 3. In total, 1,709 DEGs were identified, with 1,051 upregulated and 658 genes downregulated. The differential screening was performed according to the expression levels of protein coding genes in all samples used in the present study.

Figure 3 Volcano plot of differentially expressed genes.

Figure 4 (A-D) GO and KEGG analysis of identified DEGs in different architecture lines.

GO and KEGG analysis of identified DEGs in different architecture lines

To discern prevalent biological processes and functions among differentially expressed genes (DEGs), we conducted analyses using the Gene Ontology (GO) and the Kyoto Encyclopedia of Genes and Genomes (KEGG). The DEGs were categorized into three main GO categories: biological process, molecular function, and cellular component. The GO analysis revealed that in the biological process and cellular component category, the majority of up-regulated genes were associated with photosynthesis. In the molecular function category, the majority of genes were found to be associated with chlorophyll binding, pigment binding, and tetrapyrrole binding (Fig. 4A). Notably, a closer examination of the top GO terms revealed processes related to phytohormones, including regulation of phytohormone levels and phytohormone metabolic process in the down-regulated genes (Fig. 4B).

Furthermore, to study the biological pathways associated with the trait of leaf stalk-stem angle in different architecture lines, DEGs were annotated using blast analysis against the KEGG database. A comprehensive analysis of the KEGG pathway enrichment provided detailed insights into the top events in up and down-regulated genes. Among the up-regulated genes, major pathways associated with DEGs included photosynthesis (cmo00195), photosynthesis-antenna proteins (cmo00196), carbon fixation in photosynthetic organisms (cmo00710), glyoxylate and dicarboxylate metabolism (cmo00630) and phytohormone signal transduction (cmo04075). Conversely, down-regulated genes were associated with pathways such as zeatin biosynthesis (cmo00908), diterpenoid biosynthesis (cmo00904), glutathione metabolism (cmo00480) and plant hormone signal transduction (cmo04075). Notably, the plant hormone signal transduction pathway was enriched in both up- and down-regulated genes (Figs. 4C and 4D).

Subsequent analysis integrating transcriptome and phytohormone data revealed co-enrichment of the hormone signal transduction pathway. This observation suggests a close relationship between the melon leaf stalk-stem angles and hormonal regulation (Fig. 4E).

Differential expression pattern of phytohormone signal transduction-related genes in different architecture lines

In this section, we summarized the expression profiling of DEGs related to phytohormone signal transduction (Fig. 5). Notably, there is a distinct differential gene expression pattern observed in the signaling pathways of auxin and abscisic acid. Representative DEGs include auxin influx facilitator auxin-responsive GH3 family proteins (GH3), AUXIN/INDOLE-3-ACETIC ACID (Aux/IAA), auxin responsive factors (ARFs) and small auxin upregulated RNA (SAUR). SAUR-like auxin-responsive protein family-related DEGs were filtered and analyzed. In total, 12 DEGs associated with auxin signaling were observed (Figs. 5A–5D).

Figure 5 Transcriptomic analyses in different architecture line.

In the abscisic acid pathway, representative DEGs include abscisic acid receptor PYR/PYL family (PYR/PYL), protein phosphatase 2C (PP2C) and SNF1-related protein kinase 2 (SnRK2) family-related DEGs. In total, 5 DEGs associated with ABA signaling were observed (Figs. 5E–5G).

Discussion

The growing demand for optimal melon cultivars has intensified the focus on selecting and breeding plants with desirable traits. Leaf stalk-stem angle is a key component of the ideal plant type. Consequently, elucidating the genetic and regulatory mechanisms underlying this trait is crucial for developing melon cultivars with optimal characteristics. In addition, leaf stalk-stem angle plays a pivotal role in determining melon yield, quality and effective cultivation management. The current study delves into the variations in phytohormones and related genes in two melon lines exhibiting different leaf stalk-stem angles through physiological and transcriptomic analyses.

The branching angle of plants is a response to both gravitropic and phototropic stimuli. LAZY1 (LZY1) and related genes like LZY2 and LZY4 perceive the gravitropic signals to alter the local auxin gradient (Hill & Hollender, 2019). This asymmetric auxin distribution induces two WUSCHEL-related homeobox domain genes WOX6 and WOX11, ultimately affecting the branch/tiller angle (Zhang et al., 2018). Transcriptome data suggest the role of LZY1 in the regulation of auxin-related gene expressions, such as auxin transporters and signaling factors (Zhu et al., 2021). Therefore, auxin plays a crucial role in plant branching angle. In addition, other phytohormones are also key regulators in the development of the lamina joint and leaf angle. Brassinosteroid (BR), cytokinins (CKs), ethylene, gibberellic acid (GA), and auxin(indole-3-acetic acid [IAA]) are identified contributors in the intricate regulation of leaf angle (Huang et al., 2021; Tong & Chu, 2018). In addition, the study highlights the collaborative action of jasmonates (JAs) with other phytohormones, working synergistically to adapt the plant to the dynamically changing external environments (Jang, Yoon & Choi, 2020; Yang et al., 2019). In this study, we analyzed the phytohormones content in melon lines exhibiting different stalk-stem angles using LC-MS/MS. The result showed ACC, OPDA and IPA as significantly different in the two lines. The results demonstrated significant differences in ACC, OPDA, and IPA levels between the two lines. These phytohormones belong to the ethylene, jasmonic acid, and cytokinin classes, respectively. These findings suggest that these three phytohormones may play a crucial role in regulating melon leaf-stalk angle differentiation. However, further investigation is needed to identify the primary phytohormone responsible for this process.

This study further explores the involvement of phytohormones and signal transduction pathways in the intricate process of leaf angle differentiation, as corroborated by transcriptomic analysis. In phytohormone signal transduction pathways, different phytohormones collaborate synergistically and antagonistically to finely regulate plant morphology for better adaptation to the external environments (Dopp, Yang & Mackenzie, 2021; Fidler et al., 2022). The morphogenesis of all plants is an outcome of the intricate interaction of numerous genes and gene families orchestrated in a network (Gao et al., 2018; Li et al., 2012; Bai et al., 2012; Sakakibara, 2006). By modulating gene expression, several phytohormones assume significant roles in diverse processes (Huang et al., 2021; Tong & Chu, 2018; Luo et al., 2016). Plants adeptly maintain homeostasis and adjust to environmental changes due to the effective and systematic interaction of various phytohormones, ensuring a delicate equilibrium between growth and environmental response. In this study, 12 auxin and five abscisic acid signaling genes were identified as significantly different in the two lines. In the auxin signaling pathway, GH3 works as an auxin influx facilitator. The GH3 family genes LOC103490488 and LOC103490483 were upregulated in the melon line with large leaf stalk-stem angle. Similarly, three genes in the small auxin upregulated RNA protein family (SUAR, LOC107991561, LOC103497281 and LOC103489067) showed a similar trend as GH3. Contrastingly, two ARF genes LOC103503893 and LOC103493078 were significantly downregulated in the melon line with large leaf stalk-stem angle. These findings suggest that CH3, SUAR and ARF may play opposite roles in regulating the leaf stalk-stem angle of melon, which provides a basis for the subsequent improvement of melon plant architecture.

Within the abscisic acid pathway, five genes in the PYR/PYL, PP2C and SnRK2 pathways were identified. These genes exhibited consistent upregulation in the melon line exhibiting large leaf stalk-stem angle and downregulation in the melon line with small leaf stalk-stem angle. This result suggests that abscisic acid assumes a crucial role in the intricate process of melon leaf stalk-stem angle formation.

Conclusions

In this study, differential phytohormone profiling revealed that the levels of ACC and OPDA in the large-angled line Y164 were significantly higher than those in the small-angled line Z151, while the content of IPA in Z151 was significantly higher than that in Y164. Our findings suggest that these three plant hormones may play an important role in the differentiation process of cantaloupe. Further integration of transcriptome and hormone analyses substantiated the close relationship between melon leaf stalk-stem angle and phytohormones, especially ACC, OPDA and IPA. In summary, our study provides compelling evidence supporting the close association between phytohormones and the leaf stalk-stem angles in melon.

Supplemental Information

Supplemental Information 1 Supplement materials

Supplemental Information 2 Phytohormone data

We are grateful to Bin Liu (Xinjiang Academy of Agricultural Sciences) for his suggestions to this article.

Additional Information and Declarations

Competing Interests

Author Contributions

Data Availability

The authors declare there are no competing interests.

Jiancai Mao conceived and designed the experiments, performed the experiments, prepared figures and/or tables, authored or reviewed drafts of the article, and approved the final draft.

Haojie Wang performed the experiments, analyzed the data, prepared figures and/or tables, authored or reviewed drafts of the article, and approved the final draft.

Junhua Li performed the experiments, analyzed the data, authored or reviewed drafts of the article, and approved the final draft.

Junyan Yang performed the experiments, analyzed the data, prepared figures and/or tables, and approved the final draft.

Yongbing Zhang conceived and designed the experiments, authored or reviewed drafts of the article, and approved the final draft.

Haibo Wu conceived and designed the experiments, authored or reviewed drafts of the article, and approved the final draft.

The following information was supplied regarding data availability:

The raw sequence data are available at the Genome Sequence Archive: GSA: CRA018317.

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
