# Peer review of "Comparative transcriptome profiling suggests the role of phytohormones in leaf stalk-stem angle in melon (Cucumis melo L.)"

_PeerJ, doi:10.7717/peerj.18467_

## Round 0.1 · original submission · Major Revisions

Dear Authors

The manuscript needs a revision to be reconsidered for publication. The authors are invited to revise the paper considering all the suggestions made by the reviewers. Please note that the requested changes are required for publication.
With Thanks

Reviewer 1 ·

Basic reporting

Thank you for considering me for reviewing the manuscript “Comparative transcriptome profiling suggests the role of phytohormones in leaf stalk-stem angle in melon (Cucumis melo L.)”. The authors employed comparative transcriptome profiling between two melon lines with differing leaf stalk-stem angles (large and small angles). The study identified differentially expressed genes and analyzed phytohormone levels to understand their roles in determining leaf stalk-stem angles. The integration of transcriptome data with phytohormone analysis is a good approach to explore key genes and pathways potentially regulating this trait in melon.

Suggestion:
Conduct a thorough English language edit of the manuscript to correct grammatical errors and improve clarity. Also, consider shortening lengthy sentences to enhance readability. Summarize redundant sentences and focus on highlighting the important aspects of the study.

The abstract needs to be improved with more details on the methodology. Also, the obtained results could be improved by including specific data points e.g., fold changes in gene expression, hormone concentration differences to give a clearer picture of the findings.

Include relevant keywords that are not presented in the title and abstract to improve searchability.

The introduction needs to be organized and improved. Different parts explored the importance of leaf stalk-stem angle in melon architecture which need to be carefully summarized and organized. Restructure by reducing redundancy and simplifying sentence structure. Explore the genetic and hormonal mechanisms underlying leaf stalk-stem angle in melon. Address the gap in the understanding of melon plant architecture, focusing on the leaf stalk-stem angle, which has implications for breeding programs aimed at improving crop efficiency. Moreover, the introduction could be improved by adding more recent references, especially studies on similar traits. The knowledge gap should be emphasized, and the novelty of the study should be more highlighted. State the novelty of the research, i.e., what differentiates this study from previous research. Add references to recent studies that highlight the current trends. The hypothesis and objectives need to be clearly defined.

The methods are generally well described. The use of RNA-seq and phytohormone profiling is applicable for the research hypothesis. The phytohormone analysis could be improved by justifying selection of specific phytohormones and their relevance to the trait being studied.

The results section is well-written. It could be improved by presenting in a logical flow that aligns with the research hypotheses. For example, start with the overall differential gene expression, followed by specific gene families or pathways, and then move to phytohormone analysis and their integration. Also, the analysis of differential gene expression and the correlation between DEGs and phytohormone levels could be explored .
The resolution of Figures 2, 4, and 5 is currently inadequate, leading to unclear and pixelated images. For electronic publications, these figures must be clear and legible at 100% zoom.

The discussion section is very brief and should be improved. It should benefit from a comprehensive revision, incorporate recent literature to support statements. It could benefit from a clearer explanation of how the findings of this study could directly impact practical breeding programs. Exploring the potential applications in breeding or agricultural practices could strengthen the relevance of the study. The relationship between observed phytohormone levels and the leaf stalk-stem angle is not fully established and should be more critically discussed. It could improved by interpreting these correlations suggesting alternative explanations or future experiments to validate the findings.

There is no section of conclusion that should be added.

The references could be updated by including more recent studies, particularly those published in the last two years. Additionally, some references are cited without being fully discussed in the text.
For uniformity and accuracy, follow the journal style guide or citation requirements. Revise the journal abbreviations in the references for consistency.
Lines 395, 420, 424, …. : Scientific names should be in italics throughout the manuscript.

Experimental design

Experimental design is appropriate

Validity of the findings

The findings presented in the manuscript are valid, supported by robust data and sound statistical analysis.

Reviewer 2 ·

Basic reporting

No comment

Experimental design

The study demonstrates an acceptable experimental design.

Validity of the findings

No comment

Additional comments

The study provides a valuable contribution to the understanding of melon plant architecture. By comparing transcriptomic and hormonal profiles of two melon lines with contrasting leaf stalk-stem angles, the authors have successfully identified a strong association between phytohormones and this important agronomic trait.
-Comments and Suggestions for Authors
- Abstract
-The abstract mentions "phytohormones" but doesn't specify which ones were analyzed. While the results reveal the levels of ACC, OPDA, and IPA, it would be beneficial to include these specific hormones in the introductory statement about phytohormones.
- Consider emphasizing the most significant findings, such as the elevated levels of ACC and OPDA in the large-angled line and the potential role of genes involved in phytohormone signaling.
- Introduction
-While the introduction is well-structured, some sections could be condensed or rephrased for better clarity. For instance, the paragraph discussing phytohormones and leaf angle in other crops could be streamlined.
-The introduction clearly states the research gap in understanding the molecular mechanisms of leaf stalk-stem angle in melon. However, it might be helpful to emphasize the novelty and significance of this study in addressing this gap.
- Materials & Methods
-Mentioning the specific type of mass spectrometry (ESI+) and the specific MRM transitions for each phytohormone could be helpful.
-Consider summarizing some repetitive details. For example, instead of listing all quality control tests performed by the sequencing company, it might be sufficient to state that "library quality was assessed using standard metrics."
- Results
- The results effectively compare the two melon lines in terms of leaf stalk-stem angle, phytohormone levels, and gene expression.
- The study could delve deeper into the mechanisms by which phytohormones influence leaf stalk-stem angle. For example, exploring the specific interactions between phytohormones and their target genes could provide valuable insights.
Ensure consistency in the tense used throughout the section. For example, in the first paragraph, you might consider using the past tense for consistency with the rest of the section.
Here's a revised example of a sentence from the first paragraph:
Original: "There was a significant difference in the leaf stalk-stem angle between the expanded architecture line Y164 and the compact architecture line Z151 (Figure 1A)."
Revised: "We observed a significant difference in leaf stalk-stem angle between the expanded architecture line Y164 and the compact architecture line Z151 (Figure 1A)."
- Discussion
-While the discussion mentions the collaborative and antagonistic actions of phytohormones, more specific examples or references could be provided to illustrate these interactions.
-The discussion could further explore the integration of transcriptome and hormone data to provide a more comprehensive understanding of the regulatory mechanisms.

-The conclusions section is essential in a manuscript. It serves as the final synthesis of the research, providing a clear and concise summary of the key findings and their implications.

Reviewer 3 ·

Basic reporting

The results contain some new information. However, the current report relies on too much from the results of transcriptomic discovery. Some discussion should go more in-depth.

The importance of studying phytohormones needs further elaboration. The phytohormones signaling is largely regulated by dietary flavonoids, which has special health role for food plant like melon. The authors can search database like Web of Science with dietary flavonoids (Title) AND health role (Title) to get reference to enhance the discussion.

Though the transcriptomics revealed that some genes were important for certain functions. However, the transcriptomics alone has certain limitations. Thus, the results of transcriptomics should be integrated with other techniques, integrated metabolomics, for instance, to better reveal the science behind the adaptive response reported. The authors can search database like Web of Science with integrated metabolomics (Title) AND adaptive response (Title) to get reference to improve the discussion, better to have more contents for perspective.

The test with LC-MS approach needs further explanation and discussion. There were no very delicate and specific enough purification steps before the testing. Thus, the matrix effect should be there. The authors did not explain the effect of matrix compounds removal on improving the assay accurately. The authors can search database like matrix compounds removal (Title) AND assay (Title) to get reference to discuss it further.

Line 148: change the unit ‘rpm’ to ‘g’ (in italic)

Experimental design

appropriate and well-designed

Validity of the findings

good. Suggest to have more perspective thus more other approaches can enhance the transcriptomics results.

Additional comments

N/A

---

## Round 0.2 · accepted · Accept

Dear Authors,

I am pleased to inform you that the manuscript has improved after the last revision and can be accepted for publication.

Congratulations on accepting your manuscript, and thank you for your interest in submitting your work to PeerJ.

With Thanks

Reviewer 1 ·

Basic reporting

The authors have thoroughly addressed the previous comments in the revised manuscript, effectively incorporating the feedback to improve the quality and clarity of the work.

Experimental design

The experimental design is appropriate

Validity of the findings

The findings presented in the manuscript are valid, supported by robust data and sound statistical analysis.

Reviewer 2 ·

Basic reporting

no comment

Experimental design

no comment

Validity of the findings

no comment

Additional comments

The authors have made the changes I suggested in the last review. I recommend its publication in this journal.

Reviewer 3 ·

Basic reporting

N/A

Experimental design

N/A

Validity of the findings

N/A

Additional comments

The authors have addressed the questions quite well. The quality of the revised manuscript has been improved significantly. There are no further comments. The current version is acceptable for publication.